# Proanthocyanidin-Rich Grape Seed Extract Reduces Inflammation and Oxidative Stress and Restores Tight Junction Barrier Function in Caco-2 Colon Cells

**DOI:** 10.3390/nu12061623

**Published:** 2020-06-01

**Authors:** Rameshprabu Nallathambi, Alexander Poulev, Joshua B. Zuk, Ilya Raskin

**Affiliations:** Department of Plant Biology, School of Environmental and Biological Sciences, Rutgers University, New Brunswick, NJ 08901, USA; rameshprabun@gmail.com (R.N.); apoulev@sebs.rutgers.edu (A.P.); jbz27@sebs.rutgers.edu (J.B.Z.)

**Keywords:** polyphenol, proanthocyanidin, antioxidant, inflammation, tight junction

## Abstract

Grape polyphenols have previously been shown to improve gut health and attenuate the symptoms of metabolic syndrome; however, the mechanism of these beneficial effects is still debated. In this study, we investigated the protective effect of proanthocyanidin-rich grape seed extract (GSE) on bacterial lipopolysaccharide (LPS)-induced oxidative stress, inflammation, and barrier integrity of human Caco-2 colon cells. GSE significantly reduced the LPS-induced intracellular reactive oxygen species (ROS) production and mitochondrial superoxide production, and upregulated the expression of antioxidant enzyme genes. GSE also restored the LPS-damaged mitochondrial function by increasing mitochondrial membrane potential. In addition, GSE increased the expression of tight junction proteins in the LPS-treated Caco-2 cells, increased the expression of anti-inflammatory cytokines, and decreased pro-inflammatory cytokine gene expression. Our findings suggest that GSE exerts its beneficial effects on metabolic syndrome by scavenging intestinal ROS, thus reducing oxidative stress, increasing epithelial barrier integrity, and decreasing intestinal inflammation.

## 1. Introduction

Reactive oxygen species (ROS) play an important role in pathological processes in the gastrointestinal (GI) tract including diabetes, Crohn’s disease, ulcerative colitis, and colon cancer [1,2]. Prolonged exposure of the intestine to ROS leads to oxidative stress, lipid peroxidation, protein degradation, and damage to macromolecules [1,2,3,4,5]. Oxidative stress and resulting inflammation may progress to insulin resistance and symptoms of metabolic syndrome, such as hyperglycemia, hypertension, dyslipidemia, and central obesity [1,2]. The nuclear factor kappa B (NF-κB) cascade activated by ROS leads to the production of pro-inflammatory molecules, including cytokines and chemokines that may contribute to insulin resistance [3,4,5].

In Western societies, chronic oxidative and inflammatory stresses are more common, possibly due to the high fat content of food [6]. The high-fat diet elicits an inflammatory response by inducing pro-inflammatory cytokines and insulin resistance and by increasing uric acid and thiol group production, resulting from the antioxidant response triggered by ROS [3,7,8]. The high-fat diet increases whole-body inflammation and insulin resistance, which leads to metabolic syndrome and may also be mediated by the gut microbiome [9]. The high-fat diet was shown to increase the proportion of Gram-negative species in the gut, thus contributing to the increased production and absorption of pro-inflammatory bacterial lipopolysaccharides (LPS), leading to a low-grade metabolic endotoxemia [10]. The interaction of circulating LPS with TLR-4/CD-14 receptors may trigger an inflammatory cascade leading to insulin resistance, obesity, and diabetes [10,11]. The breakdown of intestinal barrier integrity, known as leaky gut, has been found to correlate with obesity in humans and has been linked to the inflammatory state associated with metabolic syndrome which results in a decrease in tight junction protein expression between cells of the epithelial lining of the gut [12,13,14,15]. Tight junctions between intestinal enterocytes function to block the extracellular permeation of macromolecules between epithelial cells and are composed of the proteins occludins, claudins, tricellulin, and junction adhesion molecule [12,14]. Zona occludens (ZO), cingulin, and afadin associate with the tight junction proteins intracellularly forming the tight junction plaque and act as an interface between the tight junctions and the actin cytoskeleton as well as various signaling pathways [14]. Inflammatory diseases decrease the gene and protein expression of tight junction proteins resulting in an increase in intestinal permeability, which is the source of metabolic endotoxemia associated with chronic obesity [12,15].

Dietary antioxidants may balance the redox network of the gut and reduce oxidative and inflammatory stress [16]. Polyphenols, the major group of dietary antioxidants contained in fruits, vegetables, cereals, tea, coffee, and wine, are a large and heterogeneous group of compounds characterized by hydroxylated phenyl moieties [17]. Flavonoids, one of the largest subclasses of polyphenols, are the major antioxidants in foods [18]. Grapes, which have high levels of antioxidant polyphenols, are consumed as fresh or dried fruit, juice, and wine [19]. Grape polyphenols (GP) have been reported to reduce obesity and atherosclerosis, change gut microbiota, and improve endothelial dysfunction [19,20,21,22]. The major polyphenols in red grapes, anthocyanins and proanthocyanidins, have been structurally characterized using Liquid Chromatography-Mass Spectrometry (LC-MS) [23]. However, a relatively poor bioavailability of grape polyphenols likely restricts their antioxidant function in the gut [9]. In rodent studies, 88–94% of administered radiolabeled anthocyanins and proanthocyanidins were recovered in the gastrointestinal tract and feces [24,25]. The present study tests the hypothesis that grape seed-derived polyphenols reduce oxidative stress, inflammation, and epithelial permeability in the LPS-induced Caco-2 colon model, and that this effect may be related to the attenuation of metabolic syndrome caused by dietary consumption of polyphenol-rich foods.

## 2. Materials and Methods

### 2.1. Grape Seed Extraction and Chemical Characterization

Grape seed powder (Changsha Huir Biological-Tech Crp. Ltd., Changsha, China) was extracted in 25% aqueous ethanol in a ratio of 1:5 (g:mL), i.e., 10 g of powder in 50 mL of 25% ethanol. The extraction mixture was sonicated for 25 min at room temperature, centrifuged at 1680× *g* for 10 min (model 5810R, Eppendorf, Hauppauge, NY, USA), and the supernatant filtered through miracloth. The filtered grape seed extract (GSE) produced from the initial 10 g of grape seed powder was dried yielding 125 mg dry extract and used for the experiments described below. Total polyphenols were quantified with Folin-Ciocalteau assay [26] and proanthocyanidins (PACs) were quantified by the 4-Dimethylaminocinnamaldehyde (DMAC) method [27] as described previously. Total polyphenols accounted for 36% by mass of dry GSE, and PACs accounted for 28%.

Separation and characterization of GSE components was done with an LC-MS system consisting of Dionex UltiMate 3000 UPLC including Dionex HPG-RS pump, RS autosampler, RS column compartment, and Dionex UltiMate photodiode array detector (PDA) and Q Exactive Plus orbitrap high resolution, high mass accuracy mass spectrometer (Thermo Scientific, Waltham, MA, USA), as described previously [19].

### 2.2. Cell Culture and Maintenance

Human colorectal adenocarcinoma cell line Caco-2 (ATCC^®^ HTB-37™, Manassas, VA, USA) was grown in 75 cm^2^ culture flasks with 10 mL of Dulbecco’s modified Eagle’s medium (DMEM) supplemented with 20% fetal bovine serum (FBS), L-glutamine, and antibiotic solution (penicillin and streptomycin) at 37 °C in a humidified 5% CO_2_, 95% atmosphere incubator. Caco-2 cells were differentiated by growth to 100% confluence and maintenance at confluence for 21 days with culture medium replaced every alternate day. Following differentiation 80% confluent cells were used for all assays.

### 2.3. Cell Viability Assay

Cell viability was measured by the 3-(4,5-dimethylthiazol-2-yl)-2,5-diphenyltetrazolium bromide (MTT) method [28]. Briefly, the cells were seeded into 96-well cell culture plates at 5 × 10^4^ cells/well and treated with different concentrations of GSE for 24 h. Subsequently, cells were incubated with MTT (0.5 mg/mL) for 4 h, and the generated formazan precipitate was dissolved with 50 µL DMSO. The absorbance was measured at 490 nm using a Synergy HT plate reader (Biotek, Winooski, VT, USA).

### 2.4. Determination of Intracellular Reactive Oxygen Species

Intracellular ROS levels were determined by fluorescence of 2′,7′-dichlorofluorescein (DCFH2-DA) [29]. Caco-2 cells were seeded into 24-well cell culture plates at a concentration of 5 × 10^4^ cells/well and cultured for 24 h. Wells were divided into 4 treatment groups (*n* = 3 wells per group) and treated for 24 h; a negative control group received only culture medium, a lipopolysaccharide (LPS) group received culture medium with LPS (25 μg/mL), a GSE group received culture medium with GSE (12.5 μg/mL), and a GSE + LPS group received both GSE (12.5 μg/mL) and LPS (25 μg/mL). Dosages of GSE and LPS were selected based on our experience of optimal dose consistent with published data [16,30,31,32]. Subsequently, cells were washed with PBS, incubated with 10 µL MDCFH-DA at 37 °C for 30 min, washed with PBS again and imaged with a fluorescence microscope (FSX100, Olympus, Waltham, MA, USA). Mean fluorescence intensity was quantified using ImageJ software (https://imagej.nih.gov/ij/) after background staining correction. Cellular oxidant levels were expressed as the mean DCF fluorescence intensity.

### 2.5. Determination of Mitochondrial Superoxide

The mitochondrial superoxide level was determined by fluorescence of MitoSOX Red fluorescence stain [33]. Caco-2 cells were seeded into 24-well cell culture plates at a concentration of 5 × 10^4^ cells/well and cultured for 24 h. Wells were treated for 24 h in 4 groups (*n* = 3 wells per group) as described above, then washed with PBS, incubated with 5 μM MitoSOX Red for 30 min, and then washed twice with PBS. Fluorescent images were captured using fluorescence microscopy and the results were expressed as mean fluorescence intensity and quantified using ImageJ software after background staining correction.

### 2.6. Detection of Mitochondrial Membrane Potential

Mitochondrial membrane potential (MMP) was determined with the fluorescent dye Rh-123 staining method, as reported previously [34]. Caco-2 cells were seeded into 24-well cell culture plates at a concentration of 5 × 10^4^ cells/well and cultured for 24 h. Cells were treated for 24 h according to the 4 treatment groups (*n* = 3 wells per group) described above, then cells were washed with PBS and stained with 10 μM of Rh-123 at 37 °C for 30 min in the dark. Cells were washed again with PBS and fluorescence intensity of the cells was visualized with fluorescence microscopy. The results were expressed as mean fluorescence intensity and quantified using ImageJ software after background staining correction.

### 2.7. Transepithelial Electrical Resistance (TEER) Assay in Caco-2 Cell Monolayers

Caco-2 cells were grown in transwell polycarbonate filters (diameter, 12 mm; pore size, 0.4 μm; Costar 3460, Corning Inc., Corning, NY, USA) in DMEM with 10% fetal calf serum and antibiotics (penicillin and streptomycin) and incubated at 37 °C with 5% CO_2_. The basolateral and apical compartments were filled with 1.5 and 0.5 mL of culture medium, respectively. The culture medium was changed three times per week, and cells became confluent after 21 days of culture. According to preliminary tests, cells were considered as confluent when the basal TEER was higher than 600 Ω. The apical medium was then replaced with medium containing GSE (12.5 μg/mL) with or without LPS (25 μg/mL) according to the 4 treatment groups (*n* = 3 wells per group) described above and incubated for another 24 h, and transepithelial electrical resistance (TEER) was measured for each treatment group using an ohm/voltmeter (EVOM, WPI, Sarasota, FL, USA) as previously described [35]. Resistance values were calculated in Ω·cm^2^ by multiplying the resistance values by the filter surface area.

### 2.8. Detection of Interleukins by Enzyme Linked Immunosorbent Assay (ELISA)

Caco-2 cells were seeded into 24-well cell culture plates at a concentration of 5 × 10^4^ cells/well and cultured for 24 h. Cells were treated with GSE (12.5 μg/mL) with or without LPS (25 μg/mL) for 24 h in the 4 treatment groups described above. The supernatant from each treatment was used to determine the levels of interleukin 8 (IL8) and interleukin 6 (IL6) protein released by the cells using commercial Human DuoSet ELISA kit (R&D Systems, Minneanapolis, MN, USA).

### 2.9. Immunofluorescence Microscopy

Caco-2 cells were grown on 22-mm glass coverslips in a humidified CO_2_ incubator. The cell monolayer was pretreated with GSE (12.5 μg/mL) with or without LPS (25 μg/mL) in 4 treatment groups (*n* = 3 coverslips per group) described above for 24 h and fixed with ice cold methanol. The tight junction proteins zona occludens 1 (ZO1), occludin, and claudin were immunolabelled with fluorescent polyclonal antibody (Abcam, Cambridge, MA, USA) and counter stained with Hoechst nuclear stain. After washing with PBS, the cells were imaged with confocal microscopy (LSM710 model, Zeiss, Oberkochen, Germany) and the relative fluorescence intensity was quantified using ImageJ.

### 2.10. mRNA Extraction and Gene Expression

Caco-2 cells were seeded into 24-well cell culture plates at a concentration of 5 × 10^4^ cells/well and cultured for 24 h. The cell monolayer was treated with GSE (12.5 μg/mL) with or without LPS (25 μg/mL) for 24 h in the 4 treatment groups (*n* = 3 wells per group) described above, then the cells were washed 2× in PBS, collected in TRIzol Reagent (Life Technologies, Carlsbad, CA, USA), and stored at −80 °C for RNA extraction. Gene expression assays were performed as described previously [36]. Briefly, total RNA was extracted from cells in TRIzol Reagent and treated with Deoxyribonuclease I (Life Technologies) according to the manufacturer’s instructions. RNA quality was assessed on the NanoDrop 1000 (NanoDrop Technologies, Wilmington, DE, USA) and cDNA synthesis was performed using the ABI High Capacity cDNA Reverse Transcription Kit (Applied Biosystems, Foster City, CA, USA) with RNAse I inhibitor, according to the manufacturer’s instructions, using 5 μg RNA as a template in a 25 μL reaction. For both the oxidative and inflammatory markers analysis, cDNA samples were diluted 25-fold for qRT-PCR analysis on the QuantStudio 3^®^ Real-Time PCR System (Applied Biosystems) with Power SYBR Green PCR master mix (Applied Biosystems) and primers used were as follows: Glyceraldehyde 3-phosphate dehydrogenase (GAPDH) forward 5′-GGA AGG TGA AGG TCG GAG TC-3′, reverse: 5′-TCA GCC TTG ACG GTG CCA TG-3′; tight junction protein 1 (TJP1) forward 5′-CGG GAC TGT TGG TAT TGG CTA GA-3′, reverse: 5′-GGC CAG GGC CAT AGT AAA GTT TG-3′; glutathione-disulfide reductase (GSR) forward 5′-AAG GGT CAT ATC ATC GTA G-3′, reverse: 5′-GTC TCC TGG TTC TCA ACG A-3′; superoxide dismutase 1 (SOD1) forward 5′-AGG GCA TCA TCA ATT TCG AG-3′, reverse: 5′-ACA TTG CCC AAG TCT CCA AC-3′; superoxide dismutase 2 (SOD2) forward 5′-TCC ACT GCA AGG AAC AAC AG-3′, reverse: 5′-TCT TGC TGG GAT CAT TAG GG-3′; glutathione peroxidase 2 (GPX2) forward 5′-AGT CTC AAG TAT GTC CGT C-3′, reverse: 5′-CCT TTA TTG GTC TCT TCT A-3′; interleukin 1 alpha (IL1 forward 5′-ATG GCC AAA GTT CCA GAC ATG-3′, reverse: 5′-TTG GTC TTC ATC TTG GGC AGT CAC-3′; interleukin 6 (IL6) forward 5′-CAT CCT CGA CGG CAT CTC AG-3′, reverse: 5′-GCT CTG TTG CCT GGT CCT C-3′; interleukin 10 (IL10) forward 5′-TCA GGG TGG CGA CTC TAT-3′, reverse: 5′-TGG GCT TCT TCT AAA TCG TTC-3′; tumor necrosis factor alpha (TNFα) forward 5′-TCT CGA ACC CCG AGT GAC AA-3′, reverse: 5′-TAT CTC TCA GCT CCA CGC CA-3′; transforming growth factor beta-1 (TGFβ-1) forward 5′-GCT GCT GTG GCT ACT GGT GC-3′, reverse: 5′-CAT AGA TTT CGT TGT GGG TTT C-3′. The thermal cycler profile was as follows: 2 min, 50 °C; 10 min, 95 °C; 15 s, 95 °C; 1 min, 60 °C for the dissociation stage; 15 s, 95 °C; 1 min, 60 °C; 15 s, 95 °C for 40 cycles. Gene expression was quantified by the comparative ΔΔCt method and normalized to GAPDH. The cells without any treatment served as the calibrator and were assigned a value of 1.0.

### 2.11. Statistical Analyses

All values are expressed as mean ± S.D. (*n* = 3). Data were analyzed with a one-tailed analysis of variance (ANOVA) using the GraphPad prism software and differences between means were assessed post-hoc using Tukey’s test. *p* < 0.05 was considered statistically significant.

## 3. Results

### 3.1. Polyphenol Characterization and Cytotoxicity of GSE

We first sought to characterize the polyphenols that were present in the GSE preparation used in the experiments of this study. The LC-MS spectrum analysis revealed the presence of proanthocyanidin monomers (catechin, epicatechin), dimers (procyanidin B1, B2, B3, B4, B5, B7 etc.), trimers, tetramers, pentamers, and corresponding gallates (Figure 1). GSE applied at 0, 3.13, 6.25, 12.5, 25, and 50 μg/mL had no significant cytotoxic effects (*p* > 0.05) on the Caco-2 cells after 24 h of treatment as determined by MTT assay.

### 3.2. Effect of GSE on Oxidative Stress

In order to understand the relationship between GSE antioxidants and LPS-induced oxidative stress, we examined the effect of GSE on mitochondrial dysfunction, intracellular ROS, and mitochondrial superoxide production on LPS-treated differentiated Caco-2 cells (Figure 2). GSE at a concentration of 12.5 μg/mL (containing 3.50 μg/mL PACs out of 4.50 μg/mL total polyphenols) was effective in mitigating mitochondrial damage and significantly reducing the intracellular ROS and mitochondrial superoxide in the LPS-treated cells. GSE reversed all alterations of the LPS-induced damage to levels observed in non-treated cells, increased LPS-treated mitochondrial membrane potential by 1.5-fold, and decreased LPS-induced intracellular ROS by 2-fold (Figure 2A,B). The GSE effect on mitochondrial superoxide was particularly pronounced. GSE reduced superoxide-associated fluorescence in the LPS-induced cells by 3.5-fold (Figure 2C).

### 3.3. Effect of GSE on Antioxidant Enxyme Expression

We next examined the effect of GSE on the gene expression of selected antioxidant enzymes, GSR, SOD1, SOD2, and GPX (Figure 3). The expression of these antioxidant genes was decreased by LPS, while GSE significantly elevated expression of these genes even in the presence of LPS. In the LPS-treated cells exposed to GSE, gene expression of GSR, SOD1, SOD2, and GPX was elevated by 59%, 36%, 44%, and 70% respectively when compared to cells treated with LPS alone. GSE reduced oxidative stress in Caco-2 cells by restoring mitochondrial membrane potential, reducing mitochondrial and intracellular ROS, and significantly increasing the expression of antioxidant enzymes.

### 3.4. Effect of GSE on Tight Junction Barrier

Based on previous reports that oxidative stress also causes tight junction leakage and damage in the gut epithelial wall lining by reducing the expression of the tight junction proteins [12,16], we next sought to evaluate the hypothesis that the antioxidant properties of GSE may reduce LPS-induced tight junction leakage. We measured TEER resistance across the Caco-2 cell monolayer and we found that LPS lowered the resistance value, suggesting tight junction leakage, whereas GSE increased the TEER to the level observed in the non-treated cell layer (Figure 4A). GSE treatment also restored the expression of *TJP1* mRNA, the gene that encodes ZO1, in the monolayer of Caco-2 cells (Figure 4B), supporting the restorative effect of GSE on the integrity of the gut epithelial lining.

### 3.5. Effect of GSE on Tight Junction Protein Expression

To confirm the above observations, we also examined the ability of GSE to counteract the effects of LPS on the content of the transmembrane tight junction proteins ZO1, occludin, and claudin 1 in Caco-2 monolayer with immunocytochemistry (Figure 5A–C). We found that LPS decreased the abundance of these tight junction proteins, thus compromising the integrity of the tight junction barrier. GSE treatment significantly increased the abundance of tight junction proteins localized on the cell surfaces. Although claudin 1 showed a more defuse distribution pattern, unlike the membrane localization observed with ZO1 and occludin, its overall fluorescence intensity increased in the LPS-treated cells rescued with GSE. These immunochemistry data support the restorative effect of GSE on the tight junction barrier integrity of Caco-2 cells exposed to LPS.

### 3.6. Effect of GSE on Inflammation

Because oxidative stress is often associated with inflammatory response, we examined the anti-inflammatory effect of GSE in LPS-induced Caco-2 cells. GSE significantly reduced the secretion of IL8 (Figure 6A) and IL6 (Figure 6B) from the LPS-induced Caco-2 cells, as determined by ELISA. GSE reduced gene expression of pro-inflammatory cytokines IL1α, IL6, and TNFα in these cells, with the largest effect on the reduction of the LPS-induced IL6 gene expression by 2-fold compared to LPS positive control, and GSE increased gene expression of the anti-inflammatory cytokines IL10 and TGFβ-1, which were decreased by LPS treatment, restoring their expression to the level of non-induced negative control (Figure 6C).

## 4. Discussion

Previous studies have linked grape polyphenol consumption with a reduction in both obesity and atherosclerosis, changes in gut microbiota, and improved endothelial function [16,37,38,39]. Our study revealed that GSE decreased intracellular and mitochondrial ROS and reduced damage to the mitochondrial membrane in LPS-treated differentiated Caco-2 cells. GSE also increased the expression of stress-induced antioxidant genes (GSR, SOD1, SOD2, and GPX2) in LPS-treated and untreated cells. Additionally, GSE was found to restore expression of tight junction proteins (ZO1, occludin, and claudin 1) following suppression by LPS-treatment. Finally, our study also revealed that GSE can suppress the expression of pro-inflammatory cytokines and promote the expression of anti-inflammatory cytokines following LPS-induced inflammatory response. We suggest that these effects are mediated by antioxidant polyphenols that are the most abundant compounds in GSE.

While the antioxidant capacity of GSE has previously been characterized [19,20,21], our study revealed that GSE treatment led to an increase in expression of an array of antioxidant enzymes. The elevation of expression of GSR, SOD1, SOD2, and GPX2 suggests that the ROS scavenging effects of GSE may be both direct through the antioxidant action of grape polyphenols as well as indirect through the altered expression of these antioxidant enzymes. Our results also showed that GSE increased the expression of anti-inflammatory cytokines and decreased the expression of pro-inflammatory cytokines in LPS-treated cells. GSE reduced the expression of the LPS-induced pro-inflammatory cytokines TNFα and IL1α, which have been reported to increase the permeability of the intestinal wall by diminishing the expression of tight junction proteins [40]. Tight junctions play an important role in cell-to-cell contact and maintenance of the structure and integrity of the intestinal barrier, and are essential for regulating the transport of nutrients from the gut to the bloodstream and liquids and gases from the bloodstream into the gut [37]. Additionally, the anti-inflammatory cytokines IL10 and TGFβ-1 were suppressed by LPS, which may relate to the observed increase in the pro-inflammatory cytokines IL1α, IL6, IL8, and TNFα, and the suppression of these anti-inflammatory cytokines was reversed by GSE treatment. We also observed that GSE treatment increased the expression of TGFβ-1, a multifunctional cytokine that can normalize the epithelial barrier by upregulating the expression of tight junction proteins [41,42]. The increase in TGFβ-1 expression is consistent with the observed increase in tight junction protein expression resulting from GSE treatment and may be associated with beneficial effects on intestinal barrier function.

## 5. Conclusions

In conclusion, our results indicate that the protective effects of poorly bioavailable grape polyphenols against the symptoms of metabolic syndrome may initiate in the gut, where the ROS-scavenging ability of these compounds protects the intestinal lining from LPS-induced inflammation and restores the epithelial barrier integrity. Our laboratory has previously shown that grape polyphenols can promote the growth of beneficial bacteria, such as *Akkermansia muciniphila* in the mouse gut, which have been implicated in protection against metabolic syndrome [16,43,44]. Together with available data that suggest that grape polyphenols may serve multiple functions in combating metabolic syndrome, including acting on the gut microbiota, our study revealed that polyphenol-rich grape extract may act directly on human-derived intestinal epithelial cells to decrease inflammation and increase intestinal barrier integrity.

## Figures and Tables

**Figure 1 nutrients-12-01623-f001:**
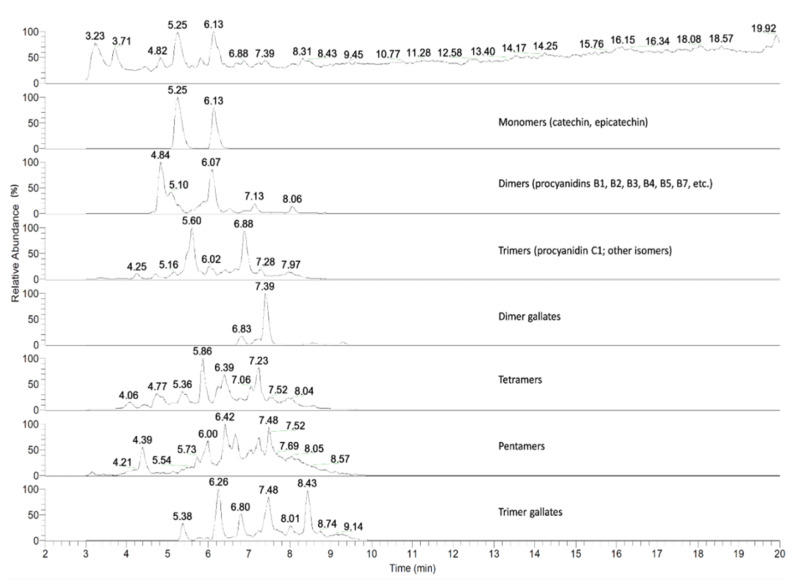
Liquid Chromatography-Mass Spectrometry (LC-MS) chromatogram of grape seed extract shows dimer, trimer, and tetramer of proanthocyanidin and its gallates. Numbers above the peaks denote the retention times.

**Figure 2 nutrients-12-01623-f002:**
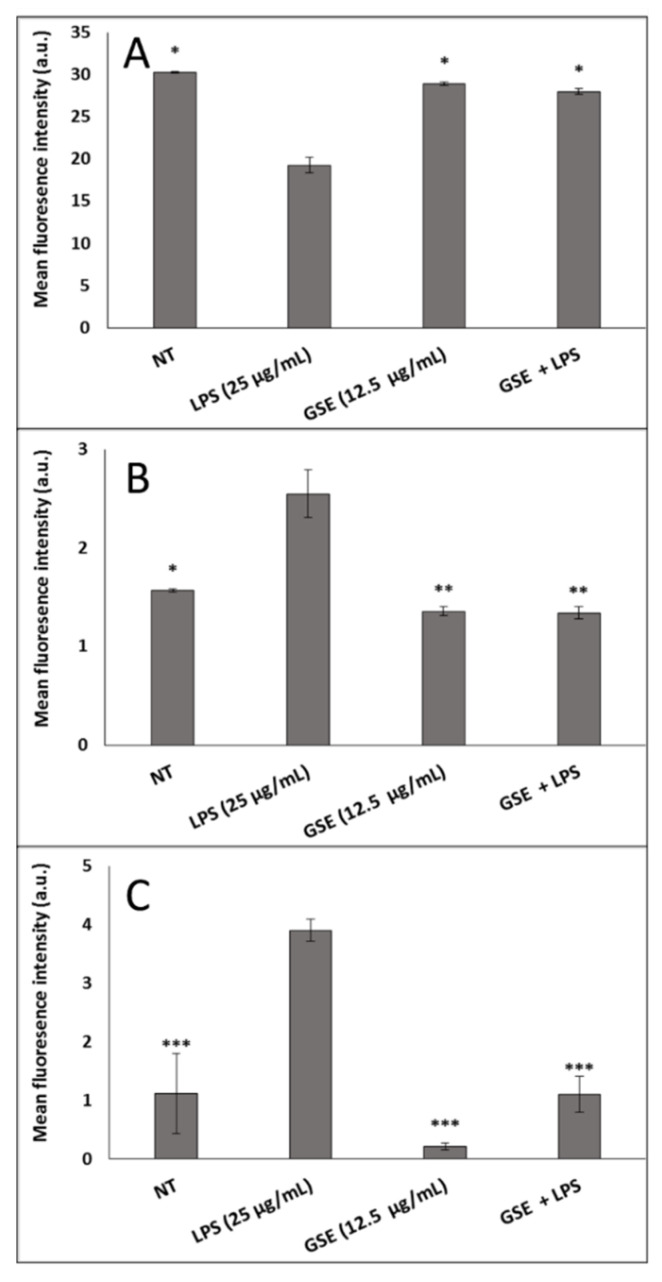
Effect of Grape Seed Extract (GSE) on mitochondrial damage in Caco-2 cells induced by Lipopolysaccharide (LPS). (**A**) After the cells were treated with LPS, GSE, and LPS + GSE for 24 h, mitochondrial membrane potential was measured revealing decreased membrane potential in LPS-treated cells compared to control while GSE treatment restored membrane potential. (**B**) Intracellular reactive oxygen species (ROS) were significantly increased in LPS-treated cells compared to control and control levels were restored by treatment with GSE as measured by 2′,7′-dichlorofluorescein (DCFH2-DA) fluorescence. (**C**) Mitochondrial superoxide was assessed with mitosox red dye revealing an increase in mitochondrial ROS in LPS-treated cells which was reduced by treatment with GSE. Error bars indicate ± S.D. (*n* = 3). *, **, and *** indicate the data were significantly different relative to LPS-treated cells with *p* ≤ 0.05, *p* ≤ 0.01, and *p* ≤ 0.001 respectively using Tukey’s test. NT, Non-Treated.

**Figure 3 nutrients-12-01623-f003:**
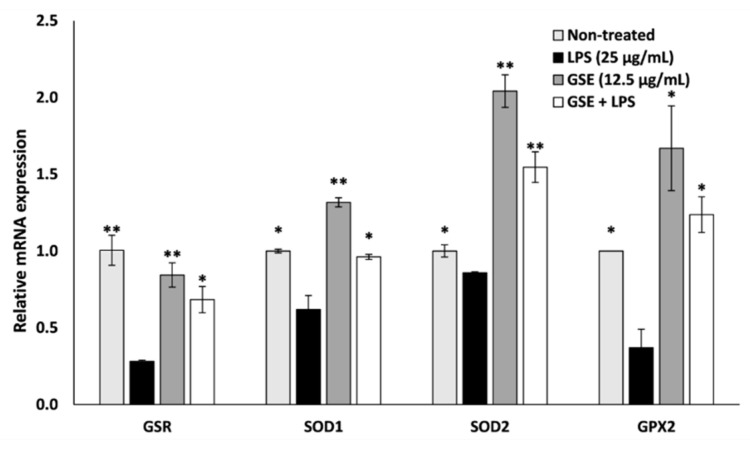
Effect of GSE on oxidative stress-related gene expression in LPS-induced Caco-2 cells. Relative expression of antioxidant genes Glutathione-Disulfide Reductase (GSR), Superoxide Dismutase 1 (SOD1), Superoxide Dismutase 2 (SOD2), and Glutathione Peroxidase 2 (GPX2) was significantly decreased in LPS-treated Caco-2 cells compared to untreated control, and expression was restored upon treatment with GSE. Error bars indicate ± S.D. (*n* = 3). * and ** indicate the data were significantly different relative to LPS-treated cells with *p* ≤ 0.05 and *p* ≤ 0.01 respectively using Tukey’s test.

**Figure 4 nutrients-12-01623-f004:**
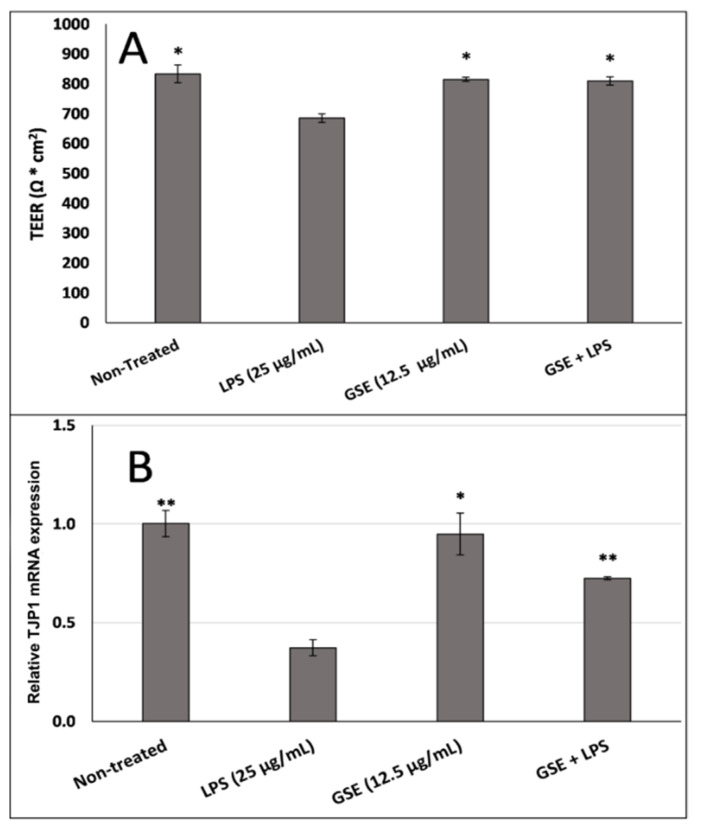
Effect of GSE on electrical resistance across (**A**) the tight junction barrier integrity measured by transepithelial electrical resistance (TEER) and (**B**) TJP1 gene expression in Caco-2 cells were significantly reduced in LPS-treated cells and restored by cotreatment with GSE. Error bars indicate ± S.D. (*n* = 3). * and ** indicate the data were significantly different relative to LPS-treated cells with *p* ≤ 0.05 and *p* ≤ 0.01 respectively using Tukey’s test.

**Figure 5 nutrients-12-01623-f005:**
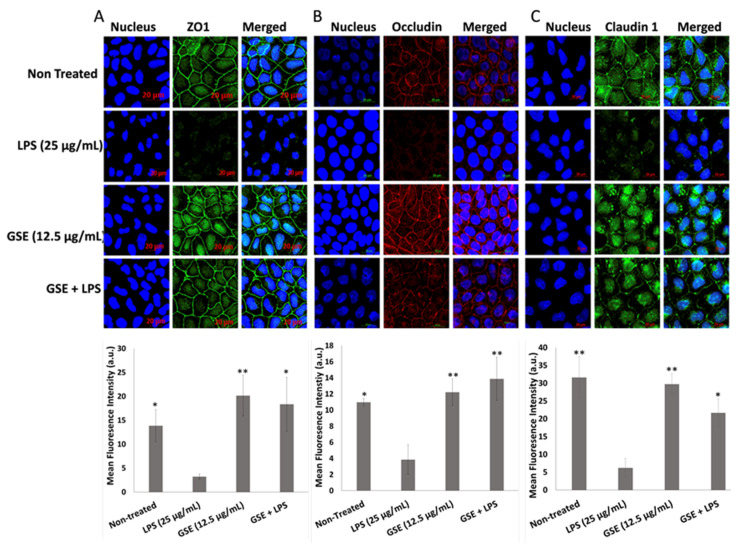
Immunolocalization (top) and its relative fluorescence intensity (bottom) of tight junction proteins. (**A**) Zona occludens 1 (ZO1), (**B**) occludin, and (**C**) claudin 1 protein expression was significantly reduced in LPS-treated cells compared to untreated control while GSE treatment significantly restored tight junction protein expression. Error bars indicate ± S.D. (*n* = 3). * and ** indicate the data were significantly different relative to LPS-treated cells with *p* ≤ 0.05 and *p* ≤ 0.01 respectively using Tukey’s test.

**Figure 6 nutrients-12-01623-f006:**
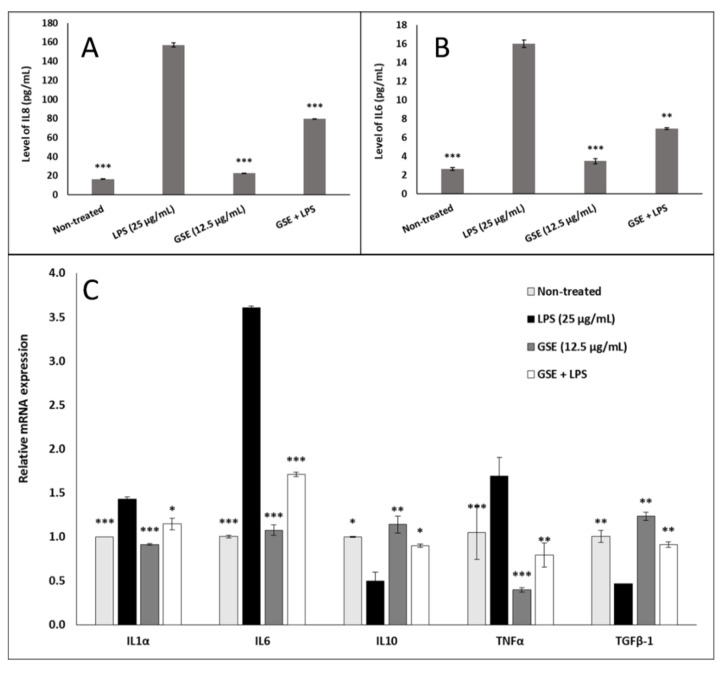
Effect of GSE on pro- and anti-inflammatory cytokines in Caco-2 cells treated with LPS, GSE, or LPS + GSE for 24 h. (**A**) Interleukin 8 (IL8) protein expression in culture medium was significantly elevated in LPS-treated cells compared to untreated control. GSE treatment alone did not significantly alter expression compared to untreated control but GSE significantly attenuated the LPS-induced increase in IL8 expression. (**B**) Interleukin 6 (IL6) protein expression in culture medium showed the same pattern as IL8. (**C**) Relative gene expression of pro-inflammatory cytokines IL1α, IL6, and tumor necrosis factor alpha (TNFα) showed a similar attenuation of LPS-induced expression upon GSE treatment. Relative gene expression of anti-inflammatory cytokines interleukin 10 (IL10) and transforming growth factor beta-1 (TGFβ-1) showed a significant decrease in LPS-treated cells compared to untreated control which was attenuated by cotreatment with GSE. Error bars indicate ± S.D. (*n* = 3). *, **, and *** indicate the data were significantly different relative to LPS-treated positive control with *p* ≤ 0.05, *p* ≤ 0.01, and *p* ≤ 0.001 respectively using Tukey’s test.

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
