# Peer review of "Proanthocyanidin-Rich Grape Seed Extract Reduces Inflammation and Oxidative Stress and Restores Tight Junction Barrier Function in Caco-2 Colon Cells"

_nutrients, 2020, doi:10.3390/nu12061623_

Round 1

Reviewer 1 Report

I have no further comments.

Author Response

Thank you.

Reviewer 2 Report

Nallathambi et al. investigated the potential beneficial effect of a grape polyphenols extract on Caco-2 function via theirs anti-inflammatory and anti-oxidant capacities. This topic is important in the obesity pandemic and in the mechanistic of polyphenols. This study is well design and the results are interesting. However, some points need developments:

Introduction.

More explanation on the proteins involved in tight junction and theirs regulation must be developed.

Methods.

Investigation of TLR4 pathway would add important information about inflammation distinctly from IL6 and 8 since the introduction is focuse on TLR4 pathway.

Why only claudin 1 was investigated since claudin 2 is frequently involved in inflammatory gut and regulated by polyphenols (Arie H, Nutrients. 2019 Nov 4;11(11):2646.doi: 10.3390/nu11112646.)

Results.

1) p6 line 206 delete “symptoms” and change by “alterations” in cell culture there is no symptoms.

2)p7 line 223 delete “suppressed by LPS” and change by “decreased by LPS”

3)Explanation and importance of TJP1 mRNA must be developed and why this protein was not investigate by immunohistochemistry or western blotting.

4)Some correlation would add important information between the functional parameter and oxidative/inflammatory markers.

Discussion.

P11 lines 310 to 312. The results do not allow such conclusion since there is no kinetics in this study.

Author Response

Reviewer 2:

Comment: “Introduction. More explanation on the proteins involved in tight junction and theirs regulation must be developed.”

Response: The introduction has been expanded to include an improved explanation of tight junction proteins and their regulation (lines 46-53).

Comment: “Methods. Investigation of TLR4 pathway would add important information about inflammation distinctly from IL6 and 8 since the introduction is focuse on TLR4 pathway.”

Response: Expression of IL6 and IL8 is a downstream effect of TLR4 pathway activation. In this study we examined the output of the TLR4 pathway rather than the entire pathway.

Comment: “Why only claudin 1 was investigated since claudin 2 is frequently involved in inflammatory gut and regulated by polyphenols (Arie H, Nutrients. 2019 Nov 4;11(11):2646.doi: 10.3390/nu11112646.)”

Response: Claudin 1 has been reported to be present in tight epithelia but absent in leaky epithelia, whereas claudin 2 has previously been reported to be present in leaky epithelia (Assimakopoulos, S. F.; Papageorgiou, I.; Charonis, A., Enterocytes' tight junctions: From molecules to diseases. World journal of gastrointestinal pathophysiology 2011, 2, 123–137.) Therefore, we reported on claudin 1 because it has an established association with non-leaky epithelia.

Comment: “Results. 1) p6 line 206 delete “symptoms” and change by “alterations” in cell culture there is no symptoms.

Response: p6 line 206 (now line 217) has been changed to “alterations.”

Comment: “2)p7 line 223 delete “suppressed by LPS” and change by “decreased by LPS””
Response: p7 line 223 (now line 236) has been updated to include this change.

Comment: “3)Explanation and importance of TJP1 mRNA must be developed and why this protein was not investigate by immunohistochemistry or western blotting.”
Response: TJP1 is the gene name for zona occludens-1 (ZO1). ZO1 protein was investigated by immunofluorescence in Figure 5. The name ZO1 has been added to the results section to eliminate the confusion (line 255).

Comment: “4)Some correlation would add important information between the functional parameter and oxidative/inflammatory markers.”

Response: NF-kB has previously been reported to mediate the inflammatory effects of ROS. The exact mechanism of our novel finding that the expression of an array of antioxidant enzymes is inversely correlated with expression of inflammatory markers was beyond the scope of this study.

Comment: “Discussion. P11 lines 310 to 312. The results do not allow such conclusion since there is no kinetics in this study.

Response: This sentence has been removed (from what are now lines 323-325).

Reviewer 3 Report

 (1)All of your data mixed with student’s t-test and Turkey's test (p<0.05) .

Thus, you have to unify the statistical methods for all of your date by only Tukey's test. Your have to change all of your figures by only Tukey's test.

You should delete student’s t-test  in 2.11. Statistical analyses.

This is very important point for evaluation of this manuscript.

(2) Figure 5 (bar graph)is not clear.

You have to improve your figure presentation of Figure 5 (bar graph).

Author Response

Reviewer 3:

Comment: “(1) All of your data mixed with student’s t-test and Turkey's test (p<0.05) . Thus, you have to unify the statistical methods for all of your date by only Tukey's test. Your have to change all of your figures by only Tukey's test. You should delete student’s t-test in 2.11. Statistical analyses. This is very important point for evaluation of this manuscript.”

Response: All data reflected in the figures were analyzed by one-tailed ANOVA followed by post-hoc Tukey’s test, which is a common and valid statistical method. The statement about the student’s t-test was in reference to preliminary data which was not included in the final version of this manuscript. Section 2.11 has been corrected to reflect the statistical method that was used for the figures in this manuscript.

Comment: “(2) Figure 5 (bar graph) is not clear. You have to improve your figure presentation of Figure 5 (bar graph).”

Response: An improved more visible version of the bar graph has been incorporated.

The authors hope that the additional revisions made to the manuscript in response to all comments by the reviewers make it suitable for publication.

Round 2

Reviewer 3 Report

There is no comments.

This manuscript is a resubmission of an earlier submission. The following is a list of the peer review reports and author responses from that submission.

Round 1

Reviewer 1 Report

This manuscript is regarding “Proanthocyanidin-rich grape seed extract reduces inflammation and oxidative stress and restores tight junction barrier function in Caco-2 colon cells”. Overall, this manuscript provides valuable findings to this fields. This manuscript need the following some major and minor revisions and professional English proof-reading.

Major comments

  1. Caco-2 cells should be differentiated into intestinal epithelial cells (21 days). For example, authors have done the cell viability assay of GSE in the non-differentiated Caco-2 cells. With differentiation, cellular morphology and its physiological characteristics are altered including a cellular toxicity of certain compounds. I think that this is a critical point using Caco-2 cells as a human intestinal epithelial model. Please explain your rationale of the study design regarding this issue. Please provide cellular toxicity result of GSE in non-differentiated Caco-2 and differentiated Caco-2.
  2. LPS concentration used in this exp, is extremely high. Have you done cellular toxicity test of LPS alone at 25 ug/mL that you used in your exp? How do you distinguish a simple toxicity versus mitochondrial damage by LPS at that high conc? What is the reason to choose 25 ug/mL LPS in this exp? Any preliminary data of dose-dependent exp of LPS?
  3. Discussion and conclusion should be improved significantly.

Minor comments

  1. Authors should include a name of city and country of manufacturers where indicated in a consistent manner. Ex. Abcam, R&D systems etc.
  2. Please provide more detailed information in the figure legend so figure itself should be self-explanatory.
  3. Figure 4, please add “TJP1 gene“ in the figure 4-B
  4. Figure 5, please make the graph readable.

Reviewer 2 Report

The manuscript entitled “Proanthocyanidin-rich grape… cells” narrates the efficacy of grape seed extract in reducing the inflammation and oxidative stress. The authors also state that grape seed extract restores tight junctions or gaps in a specific cell line. As far as my knowledge, the crude grape seed extract also shows a very strong antimicrobial and anti-fungal spectrum. Therefore, it is used by many industries as a preservative; however, the application is limited because of its taste.

I enjoyed reading the entire article; however, I didn’t find anything new in it. There are tons of papers published on grape seed extract, and on similar facts that the current paper is based on. Nevertheless, I would have appreciated it if the authors would characterize something new in grape seed extract. Proanthocyanidin is a known antioxidant compound in graphs, and several studies have been already carried out on it.

Additionally, the introduction lacks supporting references starting with the first sentence. The authors should mention exactly how much grape seed power was used and how much amount was extracted in this study.

Why 12.5µg/ml contraction? when 0-50 µg/ml showed no significant cytotoxic effects.

The graph attached to figure 5 is of low resolution.

Results and discussion should be elaborated.

Overall, the manuscript is well written, however, language needs more polishing. There are many broken sentences. Besides, authors should rewrite some sections- likewise section 2.5/2.6/2.7/2.10.

Reviewer 3 Report

The manuscript entitled "Proanthocyanidin-rich grape seed extract reduces inflammation and oxidative stress and restores tight junction barrier function in Caco-2 colon cells" submitted by Raskin, I. and co-workers describes the beneficial effects of proanthocyanidin-rich grape seed extract on metabolic syndrome by scavenging intestinal ROS. The global idea of the study presented was interesting but, in my opinion, the results are not convincing. The results presented are often very close to those obtained with non-treated. The methodology for what was presented seems adapted.

The authors also talked about that the poor bioavailability of grape polyphenols which is already reported. What is the bioavailability of GSE? Since PAs in general are very high molecular weight molecules, it is unlikely that they are absorbed as intact native forms. Contrary results were reported in the literature from studies finding neither parent PAs, nor conjugates or monomers in rat plasma or urine upon administration of a diet supplemented with dimeric and trimeric PAs or GSE. The evaluation of the half-life time on liver microsomes of grape polyphenols versus GSE would be an advantage for the paper.